# SEQUENCE OF EXPERT: BOOSTING IMITATION PLANNERS FOR AUTONOMOUS DRIVING THROUGH TEMPORAL ALTERNATION

## ABSTRACT

Imitation learning (IL) has emerged as a central paradigm in autonomous driving. While IL excels in matching expert behavior in open-loop settings by minimizing per-step prediction errors, its performance degrades unexpectedly in closed-loop due to the gradual accumulation of small, often imperceptible errors over time. Over successive planning cycles, these errors compound, potentially resulting in severe failures. Current research efforts predominantly rely on increasingly sophisticated network architectures or high-fidelity training datasets to enhance the robustness of IL planners against error accumulation, focusing on the state-level robustness at a single time point. However, autonomous driving is inherently a continuous-time process, and leveraging the temporal scale to enhance robustness may provide a new perspective for addressing this issue. To this end, we propose a method termed Sequence of Experts (SoE)—a temporal alternation policy that enhances closed-loop performance without increasing model size or data requirements. The key idea is to retain intermediate models from training that possess inherent differences in driving errors, and then alternate the activation of different models at certain temporal intervals. This approach not only preserves the consistency capability across multiple models but also leverages their differences to enhance robustness. As a plug-and-play solution for existing IL planners, our approach requires no architectural modifications or prior knowledge of scenarios, making it highly practical for real-world deployment. Our experiments on large-scale autonomous driving benchmarks nuPlan demonstrate that SoE method consistently and significantly improves the performance of all the evaluated models, and achieves state-of-the-art performance. This module may provide a key and widely applicable support for improving the training efficiency of autonomous driving models.[1]

## 1 INTRODUCTION

In the field of autonomous driving, imitation learning has emerged as one of the most important technical paradigms and the mainstream industrial solution (Chen et al., 2024), owing to its advantages of large-scale data availability, low annotation requirements, superior generalization, and driving smoothness compared to rule-based approaches. Representative efforts include Tesla's Full Self-Driving (FSD) system (Tesla, 2025) and academic works such as UniAD (Hu et al., 2023b).

However, a critical challenge arises when these IL-based planners are deployed in closed-loop environments: the problem of accumulated errors, which undermines their reliability and safety in real-world operations. In closed-loop deployment, the planner's output at each time step directly influences the vehicle's subsequent state, creating a state-action chain where the next observation depends on the previous action. As the planner's actions deviate from expert behavior, the subsequent observations drift from the training distribution, leading the model to make increasingly suboptimal decisions. The root causes of this problem are multifaceted, including distribution shift, causal confusion, etc., as concluded in (Le Mero et al., 2022; Zare et al., 2024)

---

[1]Project Code: `HidedforDouble-blindSubmission`.

Addressing accumulated errors is paramount for advancing IL-based planners toward practical deployment. Researchers have proposed various approaches to enhance the robustness of IL planners against accumulated errors. One potential solution is to continuously scale up model size or design more sophisticated architectures, allowing the model to capture the causality of the driving maneuver better and to avoid mode collapse. Representative works include (Cheng et al., 2024b; Zheng et al., 2025). Another solution is to improve the quality of training data, such as adding noise to training data, synthesizing failure data, and enhancing data fidelity (Bansal et al., 2018; Xu et al., 2025; Guan et al., 2024). Both approaches can effectively improve algorithmic performance, but they inevitably increase inference latency and data requirements.

In this work, we instead explore whether it is possible to enhance performance without increasing inference latency or data demands, by leveraging the intrinsic differences among models in error accumulation from a temporal perspective. In other words, rather than relying solely on improving the performance of a single model, we aim to improve overall autonomous driving performance through model combination, so-called Sequence of Expert.

Our method takes advantage of the inherent differences in how models accumulate and eliminate errors, treating each model as an "expert" for specific cases. Concretely, for a given model architecture and dataset, although the final training loss values across different runs may not differ significantly, we find that the representational limits of neural networks often lead each model to perform relatively better in certain types of scenarios in closed-loop driving. By combining them, we can exploit the complementary strengths of multiple models to cover a broader range of driving cases. However, this approach poses a major challenge: identifying which model excels in which scenario is itself difficult. To address this, we exploit two intrinsic properties of autonomous driving: the temporal continuity of driving and the behavior-following tendency of driving models. We design a temporal combination strategy that integrates models over time, enabling more efficient performance gains without requiring prior knowledge. This work shows the details of this design in Section 4. Extensive experiments on the closed-loop nuPlan autonomous driving benchmark demonstrate that our approach achieves state-of-the-art performance with a smaller model size, lower inference latency, and reduced training cost. We further validate the effectiveness of this framework when applied to other model architectures. The main contributions of this work are:

- This work proposes Sequence of Expert, a simple plug-and-play solution for IL-based planners. It significantly improves their closed-loop driving performance without additional computational overhead at test time.
- Through extensive evaluation, we show that SoE brings consistent performance boost to nearly all IL planners. We achieve state-of-the-art performance on the nuPlan dataset using only SoE with a baseline planner.
- We provide the optimal hyperparameter setting of SoE and a primary conjecture on why SoE works through investigative experiments.

## 2 RELATED WORKS

**Imitation Learning-based Planners**   Imitation learning-based planners leverage large-scale expert driving data to mimic human driving behaviors, while they suffer from error accumulation in real-world deployment. Recent planners incorporate techniques from the aspects of model design, data quality, or post-processing to enhance robustness in closed-loop driving. Early IL models adopt the end-to-end architecture with a single neural network (Bansal et al., 2018; Bojarski et al., 2016) while mid-to-mid (Cheng et al., 2024b) and modularized end-to-end models (Hu et al., 2023b) prevail in recent works. The corresponding scenario representation also evolves from multi-channel images encoded by convolutional neural networks (Codevilla et al., 2019) to heterogeneous information like multi-sensors (Chitta et al., 2022; Jia et al., 2023), vector (Gao et al., 2020), and graph (Wu et al., 2024), merged by the attention algorithm. Early works directly regress the action or path (Bansal et al., 2018; Codevilla et al., 2019) while recent works leverage Gaussian mixture model (Salzmann et al., 2020), anchor-based attention decoder (Shi et al., 2024), mixture of expert (Sun et al., 2024), or diffusion models (Liao et al., 2025) to learn the complex multi-modal driving behaviors. To improve the quality of training data, a typical approach is synthesizing perturbed (Cheng et al., 2024b) or failed driving data (Bansal et al., 2018) not presented in the expert training set so that IL planners can learn how to recover from those states. Researchers on world models (Guan et al., 2024) are

trying to generate real-level driving scenarios or rare happened corner cases and using them to train IL planners. Fallback strategies are proposed and used in the downstream of IL planners, serving as an extra dimension to address the accumulated errors that IL planners fail to handle. These strategies are usually handcrafted rules, using convex optimization (Hu et al., 2023b;a) or scoring (Cheng et al., 2024a) to ensure smooth and safe trajectories. Rather than relying solely on improving the performance of a single model, this work aims to enhance IL planners from a new perspective. We design a temporal combination strategy that integrates models over time, enabling more efficient performance gains without modification to the model and data.

**Ensemble model**    Ensemble method leverages multiple learning algorithms to obtain at least two high-bias and high-variance models to be combined into a better-performing model (Rokach, 2010). It's usually used in classification (Iqball & Wani, 2023; Pham et al., 2021) and regression (Qiu et al., 2014; Van Hasselt et al., 2016) tasks, where the results from base learners are aggregated by weighted averaging, selection algorithm, or voting. In the field of autonomous driving, researchers in (Wu et al., 2022) propose a model that has two branches for trajectory planning and direct control, which are then fused with prior knowledge of driving scenarios. Although the SoE is similar to the ensemble method in that both use multiple models, they have several distinct features. Ensemble methods run all sub-models per inference step, while SoE only runs a single model, which doesn't bring any computational overhead. In contrast to classification and regression, there is no straightforward motivation or method for aggregating paths from different models in the planning task. Based on the temporal continuity of driving, SoE realizes the aggregation of the driving skills of models in the temporal dimension.

**Mixture of Expert**    Mixture of Expert (MoE) is a widely adopted paradigm in machine learning (Fedus et al., 2022; Dai et al., 2024; Sun et al., 2024) that leverages multiple specialized sub-models (experts) alongside a gating mechanism to dynamically route inputs to the most relevant components. Originally proposed in (Jacobs et al., 1991), MoE architectures improve model capacity and computational efficiency by enabling conditional computation. SoE is also similar to MoE from the perspective that it uses multiple specialized experts to improve model performance. In Appendix A, we show supporting materials that the models in SoE specialize in different driving scenarios. Both methods also have distinct features. Sub-models are nested within the main model and trained together in MoE, while models in SoE are independent and trained separately. The "experts" in MoE work in parallel, scheduled by the router, while those in SoE are sequential and scheduled by rule.

## 3 PRELIMINARIES

### 3.1 PROBLEM DEFINITION

We formulate autonomous driving as a Markov Decision Process (MDP) $\mathcal{M} = (\mathcal{S}, \mathcal{A}, P, r)$, where $\mathcal{S}$ denotes the state space (e.g., ego-vehicle and surrounding agents), $\mathcal{A}$ the action space (e.g., planned trajectory or control commands), $P : \mathcal{S} \times \mathcal{A} \to \Delta(\mathcal{S})$ the transition dynamics, and $r : \mathcal{S} \times \mathcal{A} \to \mathbb{R}$ a reward function measuring safety, progress, and comfort.

In imitation learning (IL), the reward function is not directly accessible. Instead, a policy $\pi(a \mid s)$ is trained to mimic expert demonstrations $\mathcal{D} = \{(s_i, a_i)\}$ by minimizing the supervised imitation loss:

$$\min_{\pi} \ \mathbb{E}_{(s,a)\sim\mathcal{D}}\big[\ell(\pi(a \mid s), a)\big]. \tag{1}$$

While this ensures good open-loop (OL) accuracy, the deployment objective is the closed-loop (CL) return:

$$J(\pi) = \mathbb{E}_{\tau\sim\pi,P}\Big[ \sum_{t=0}^{T} r(s_t, a_t) \Big], \tag{2}$$

where the trajectory $\tau = (s_0, a_0, \dots, s_T, a_T)$ is generated by interaction with the environment.

### 3.2 LIMITATIONS OF IL PLANNERS

A critical limitation of imitation learning (IL)-based planners is the phenomenon of *error accumulation*. In closed-loop (CL) deployment, each predicted action directly influences the next state. Even

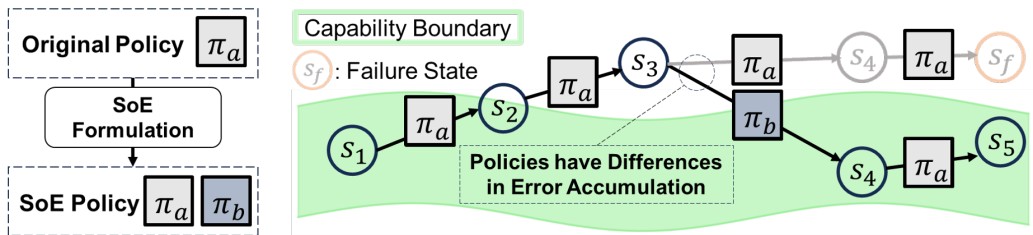

Figure 1: SoE Framework creates a SoE policy from any IL planners. One policy $\pi_a$ may drive the next state out of its capability boundary, leading the state trajectory to a failure state. We find that the component policies have natural differences in error accumulation. The insight is to use another expert $\pi_b$ to help it back to its capability coverage in the state space.

small deviations from expert trajectories can gradually shift the vehicle into states unseen during training, causing the policy to operate outside its capability boundary. As a result, errors compound over time and eventually lead to cascading failures such as collisions or off-road maneuvers (Le Mero et al., 2022; Dauner et al., 2023). This explains why minimizing open-loop (OL) imitation loss does not necessarily translate to robust CL driving performance.

Motivated by this gap, we conducted an empirical study across multiple IL planners and training runs. Our analysis revealed several consistent phenomena that highlight structural weaknesses of current approaches and provide new opportunities for improvement. Supporting evidence for these findings is given in Appendix A, and they are summarized as follows:

- **OL–CL mismatch.** The planners achieving the best OL imitation accuracy are often not those that deliver the best CL performance, a discrepancy also reported in (Dauner et al., 2023). While OL metrics improve steadily with training, CL metrics fluctuate significantly.
- **Early-stage CL superiority.** The highest CL scores often arise from earlier training epochs rather than from the final converged models. This effect is more pronounced when the deployment distribution diverges strongly from the training distribution.
- **Inter-run complementarity.** Models trained with identical architectures and data but different random seeds achieve nearly indistinguishable OL losses, yet their CL performance differs substantially. We observed that these models exhibit complementary strengths in handling diverse failure cases. This suggests that training variance is not noise but an exploitable source of diversity.

Together, these observations reveal that IL planners are inherently limited when relying on a single model. They directly motivate our proposed *Sequence of Experts (SoE)* framework, which leverages inter-model complementarity to alleviate error accumulation in a plug-and-play manner.

## 4 METHODOLOGY

### 4.1 OVERVIEW

The closed-loop deployment of imitation learning (IL) planners inevitably faces the problem of *error accumulation*: small inaccuracies at early steps compound over time, eventually driving the system into states beyond a single model's capability. Our key insight is that such failures need not be handled by the same policy throughout execution. Instead, another expert policy $\pi_b$ can temporarily take over when one policy $\pi_a$ drifts toward failure states, as illustrated in Fig. 1.

This perspective raises two essential questions: (i) *when* should an expert engage, and (ii) *which* expert should be selected. The former concerns identifying moments where $\pi_a$ is prone to failure, while the latter requires another expert $\pi_b$ that is both similar enough to $\pi_a$ to avoid introducing new weaknesses yet complementary enough to cover the gaps in $\pi_a$'s capability. Our framework addresses these challenges by systematically designing a temporal sequencing mechanism for expert engagement.

Unlike ensemble learning, which aggregates multiple models in parallel, or mixture-of-experts, which introduces additional routing complexity, our approach explores a new axis of model composition:

*temporal expert sequencing*. This design requires no architectural modifications, no extra inference cost, and can be seamlessly integrated with existing IL planners, making it both practical and widely applicable.

## 4.2 SEQUENCE OF EXPERTS (SOE) FRAMEWORK

**SOE Task Definition**    Let $\mathcal{N} = \{\pi_1, \pi_2, \ldots, \pi_n\}$ denote a set of IL policies obtained by training the same architecture with different random seeds on the same dataset. At each timestep $t$, a scheduling function $\sigma : \mathbb{N} \to \{1, \ldots, n\}$ determines which expert policy to invoke. The resulting SoE policy is defined as

$$\pi_{\text{SoE}}(a_t \mid s_t, t) = \pi_{\sigma(t)}(a_t \mid s_t), \quad \pi_{\sigma(t)} \in \mathcal{N}. \tag{3}$$

In this formulation, SoE transforms model variance into a systematic source of robustness. By alternating across experts in the temporal domain, SoE reduces the likelihood of persistent error accumulation from a single model. This perspective complements existing approaches: whereas ensemble learning aggregates predictions in parallel and mixture-of-experts dynamically routes inputs, SoE composes policies sequentially in time, thereby opening a new axis for exploiting expert diversity in IL planning.

**Temporal Scheduling**    During the inference of policy, finding the states that could lead to policy's failure requires a comprehensive prediction of future states. Although there are methods like Monte Carlo Tree Search (Silver et al., 2017) navigating decision space, the state space of traffic scenarios is still too large to be completely considered. This work proposes what may be the simplest scheduling strategy, where $\mathcal{N} = \{\pi_a, \pi_b\}$ contains only 2 expert policies and the second expert policy engages the planning process periodically. The scheduling function is as follows:

$$\sigma(t) = \begin{cases} a, & t\%n < n - 1 \\ b, & t\%n = n - 1, \end{cases} \tag{4}$$

where $n$ is an integer hyperparameter representing the period, and t denotes the discrete time step index starting from 0. This combination strategy doesn't require any prior knowledge of driving scenarios, making it highly practical for real-world deployment. Despite the simplicity of the scheduling strategy, experiment results show that it yields significant improvements in policy performance, further demonstrating the potential of the SoE framework. It's worth noting that bypassing this problem with periodical engagement brings additional challenges to the selection of expert policy. Now that another expert policy engages more frequently, it has a greater chance of introducing errors that the original policy can't handle, resulting in severe failure.

**Expert Selection**    This work proposes to use the same model trained with the same data but a different seed as the expert policy. The idea arises from the key phenomena observed during the training of IL planners. We noticed that the models across training exhibit great variance in the closed-loop performance, and this observation exists for nearly all planners. The models trained with different seeds have the same architecture and training data, which implies their natural similarity, making them the perfect expert policy for themselves.

This phenomenon, where different random seeds lead to qualitatively different behavior under distribution shift, is also reported in (Jordan, 2024; Madhyastha & Jain, 2019). This variability stems from the non-convex loss surface inherent in neural network training. Neural networks are optimized via stochastic gradient-based methods. Consequently, different random initializations, data order, and other factors can steer the training process toward distinct local minima in the weight space. While these minima may perform similarly on the training data, they often diverge in their generalization capabilities, particularly when faced with an out-of-distribution shift. Researchers in (Jordan, 2024) claimed that such variance between independent runs is unavoidable, even with identical hyperparameters, a finding that is consistent with our own observations.

## 4.3 ALGORITHMIC IMPLEMENTATION

**Pipeline**    With both problems solved, the complete steps to formulate the SoE policy are concluded in Algorithm.1 and as follows:

---

**Algorithm 1:** Sequence of Experts (SoE) Construction

---

**Input:** training dataset $\mathcal{D}$, base architecture $A$, number of runs $m$, scheduling period $n$
**Output:** SoE policy $\pi_{\text{SoE}}$
**for** $i = 1$ *to* $m$ **do**
  Train policy $\pi_i$ on $(A, \mathcal{D})$ with random seed $i$;
  Select best checkpoint $\pi_i^*$ on validation set;
$\mathcal{M} \leftarrow \{\pi_1^*, \pi_2^*, \ldots, \pi_m^*\}$;
Construct SoE policy $\pi_{\text{SoE}}$ with all possible $\mathcal{N} \in \mathcal{M}$ and temporal scheduling function $\sigma$;
Evaluate $\pi_{\text{SoE}}$ on validation set;
**return** best-performing $\pi_{\text{SoE}}$;

---

1. Training model $m$ times with different seeds and saving a checkpoint for each epoch.

2. For each training, select the top performance checkpoint among the epochs with the validation set $\mathcal{V}$. $\mathcal{V}$ should stand for the target set where the model is expected to work. This step will produce a model set $\mathcal{M}$ containing $m$ models.

3. For all the possible binary combinations $\mathcal{N} \in \mathcal{M}$, combining both as the SoE policy and testing them with $\mathcal{V}$. Selecting the best SoE policy as the final policy. While a larger set of experts, e.g., $|\mathcal{N}| = 3$, is feasible, it necessitates the use of different scheduling functions.

**Hyperparameter** SoE introduces only two parameters: the scheduling period $n$ and the number of trainings $m$, making it highly user-friendly. Furthermore, as will be demonstrated in the next section, $n$ has a practically optimal value of 2.

## 5 EVALUATION

### 5.1 EVALUATION SETTING

**nuPlan** nuPlan is a large-scale real-world planning benchmark for autonomous driving. Through its simulation framework, it introduces three simulation modes: open-loop (OL), closed-loop non-reactive (CL-NR), and closed-loop reactive (CL-R). In open-loop mode, the ego vehicle is controlled via log-replay, whereas in closed-loop modes, control is managed by a planning model. Similarly, traffic agents are controlled by log-replay in non-reactive mode and by the traffic model in reactive mode, where the traffic model is the Intelligent Driver Model (Treiber et al., 2000). For evaluation, the typical approach in the community is to use Val14 (Dauner et al., 2023) as the validation set and the nuPlan online benchmark as the final test set. However, since the online benchmarks are currently unavailable, and given the claim in (Dauner et al., 2023) that Val14 yields results nearly identical to the online benchmark, current researchers often utilize Val-14 for both model selection (validation) and final reporting (testing). To ensure a fair comparison, we adopt this common setting and also report results on the Test14 split. However, this common practice introduces a data leakage problem. To address this, we propose the Train14 split as additional validation set. Train14 and Val14 contain the same number and type of scenarios, but all scenarios within Train14 are sourced exclusively from the original training data. This ensures zero scenario overlap between the two sets. The scenarios were not manually selected but were generated in a one-shot process via random number generation. The detailed description of all splits are in Appendix B.

**Metrics** The metrics differ in open-loop mode and closed-loop mode. In the open-loop test, the metrics consist of imitation scores, measuring the similarity between the planner's output trajectory and the expert trajectory. The metrics of closed-loop testing try to quantify the real driving performance, including multiplied scores that measure severe failure and additive scores that measure efficiency and comfort. The concrete explanations of each metric can be found in Appendix B. All metrics are percentages, where higher is better.

Based on nuPlan's metrics, we define two indicators to quantify the improvements from the SoE. The average improvement $\lambda$ is the difference between the average performance of all possible SoE policies and the average of the individual expert policies, calculated as $\lambda = \overline{f(\pi_{SoE})} - \overline{f(\pi_a)}$,

where $f()$ denotes nuPlan's test score of a policy. The policy improvement $\theta$ is expressed as $\theta = f(\pi_{SoE}) - \max(f(\pi_a), f(\pi_b))$, quantifying the improvement from a SoE policy over its components.

**Baselines** The baseline planners can be classified into three groups: rule-based planners, learning-based planners, and hybrid planners. The main body of learning-based planners and hybrid planners consists of neural networks trained to imitate experts' driving, while hybrid planners incorporate additional human-defined rules to enhance performance. We construct the SoE policy for all planners except the rule-based ones. The parentheses following the name denote (Model Parameters, Training Samples, Training Epochs), where m denotes million. The detailed description of planners is as follows:

- PDM series(Dauner et al., 2023)(1m, 0.15m, 100): PDM-Closed is a rule-based planner that selects a centerline, forecasts the environment, and creates varying trajectory proposals. PDM-Open is a multi-layer perceptron whose inputs are only the centerline extracted by IDM and the ego history.
- Raster model(Caesar et al., 2021)(20m, 1m, 25): A planner takes multi-channel images of ego-centered driving scenarios as input and regresses a single trajectory as output.
- PlanTF(Cheng et al., 2024b)(2m, 1m, 25): A planner built on a transformer architecture. It encodes vectorized scenarios and regresses multiple trajectories in one forward pass.
- Pluto(Cheng et al., 2024a)(4m, 1m, 25): An improved version of PlanTF, incorporating contrastive learning and trajectory queries to enhance the model performance.
- DiffusionPlanner(Zheng et al., 2025)(6m, 1m, 500): A state-of-the-art planning model based on diffusion transformer (Peebles & Xie, 2023) which can effectively model multi-modal driving behavior. In particular, we omit the SoE construction of it due to limited computing resources and leave it for future work.

**SoE Setting** Unless otherwise stated, the setting of SoE remains the same for all the experiments. The number of trainings $m$ is 4 and the period $n$ is 2. All the results are one-shot without additional training and selection.

## 5.2 RESULTS

**Quantitative Results** Table 1 shows the closed-loop performance of baselines and their SoE version. The open-loop performance is omitted because SoE does not influence it. First of all, SoE brings a significant and latency free performance boost to all IL planners, regardless of their model architectures, except for PDM-Open. It's normal because PDM-Open's inputs only contain the center line and ego history, which are tailored for OL testing and not strictly a planner. SoE also applies to IL planners with post-processing, i.e., Pluto, achieving a remarkable performance improvement without any modification to its rules. In particular, the SoE version of both Pluto without rule and Pluto achieves state-of-the-art performance in their respective categories, requiring less inference cost and training cost compared to the strong baseline, DiffusionPlanner.

Table 2a enumerates the Val14 CL-NR scores of all SoE combinations of Pluto w/o. The diagonal elements whose $\pi_a$ and $\pi_b$ are the same are the scores of the four individual expert policies, and the rest are the scores of SoE policies. The average improvement $\lambda$ is then 0.85%, which is a significant improvement considering that there is only around 10% left to reach 100%. These results support the universal effectiveness of SoE as a plug-and-play solution for IL planners.

**Qualitative Results** Fig.2 lists the frames from one driving log of planTF and its SoE version. The period $n$ is 2, so the even-numbered frames are from the expert policy $\pi_a$ and the odd-numbered frames are from the expert policy $\pi_b$. As we can see in the upper row, $\pi_a$ drives the ego crashing into a parked vehicle. In frame 46, the states are nearly identical in both frames, which shows that the $\pi_b$ didn't cause a significant deviation from the original policy $\pi_a$. In frames 51 and 52, a clear divergence is evident between the two policies. $\pi_b$ generates a stopping trajectory while $\pi_a$ keep neglecting the parking vehicle. In frames 61 and 62, both policies reach a consensus and generate parking trajectories. The SoE policy finally stops the vehicle in frame 90. This example shows how SoE improves the performance of IL planners. $\pi_b$ interrupts the error accumulation of $\pi_a$, avoiding the collision that $\pi_a$ would have caused.

Table 1: Closed-loop performance of baselines and their SoE version. w/o are planners without fallback strategies. SoE1 and SoE2 denote two different instances of SoE combinations. Planners are classified by (R)ule, (L)earning and (H)ybrid. Val14 used for both validation and test. More implementation details about these results are described in Appendix B

| Type | Planner | Val14 (%) | | Test14 (%) | | Time (ms) |
|------|---------|-----------|-----------|------------|-----------|-----------|
| | | CL-NR | CL-R | CL-NR | CL-R | |
| R | Log-Replay | 93.53 | 80.32 | 94.03 | 75.86 | - |
| | IDM | 75.60 | 77.33 | 70.39 | 72.42 | 41 |
| | PDM-Closed | 92.84 | 92.13 | 90.05 | 91.64 | 122 |
| L | PDM-Open | 53.64 | 56.89 | 52.91 | 58.10 | 65 |
| | PDM-Open-SoE | 54.09 (+0.45) | 56.73 (-0.16) | 52.75(-0.16) | 59.85(+1.75) | 65 |
| | Raster | 70.77 | 69.07 | 65.87 | 69.93 | 37 |
| | Raster-SoE | 72.05 (+1.28) | 71.47 (+2.40) | 69.89(+4.02) | 72.86(+2.93) | 37 |
| | planTF | 84.69 | 76.15 | 85.60 | 78.86 | 78 |
| | planTF-SoE | 86.90 (+2.07) | 79.70 (+2.92) | 88.06(+2.46) | 78.88(+0.02) | 78 |
| | Pluto w/o | 88.78 | 78.01 | 90.67 | 78.06 | 107 |
| | Pluto w/o-SoE1 | **90.94** (+2.16) | 81.81 (+3.80) | **91.39**(+0.72) | 82.29(+4.23) | 107 |
| | Pluto w/o-SoE2 | 90.04 (+1.26) | 82.36 (+4.35) | 90.85(+0.18) | **83.07**(+5.01) | 107 |
| | Diffusion w/o | 89.85 | **82.88** | 89.40 | 82.67 | 159 |
| H | Pluto | 92.79 | 89.77 | 92.55 | 90.29 | 259 |
| | Pluto-SoE | **94.56** (+1.77) | 91.17 (+1.40) | **94.82** (+2.27) | 90.47 (+0.18) | 259 |
| | Diffusion | 94.26 | **92.90** | 94.80 | **91.75** | - |

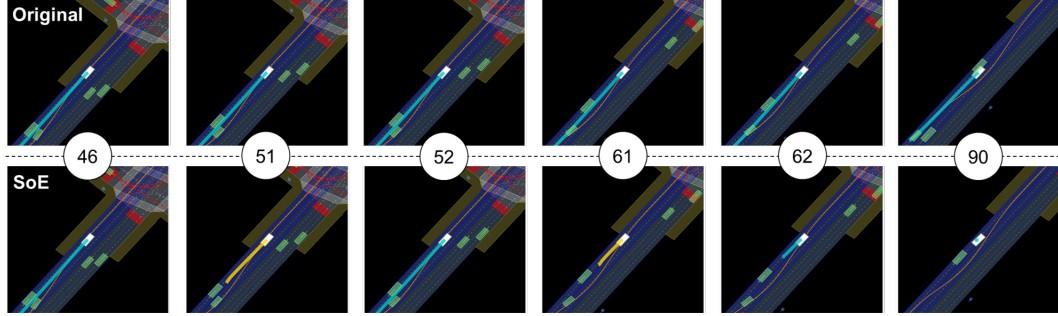

Figure 2: One driving log of planTF(upper row) and its SoE version(lower row). The number indicates the timestamps of bird-view images. The cyan trajectory is from the original expert policy $\pi_a$ and the gold one is from another expert policy $\pi_b$.

Table 2: The complete Val14 CL-NR scores of Pluto w/o's SoE combinations (a) and ablation combinations (b).

| $\pi_a$ \ $\pi_b$ | $m_1$ | $m_2$ | $m_3$ | $m_4$ |
|------|-------|-------|-------|-------|
| $m_1$ | 89.43 | 90.90 | 89.84 | 89.36 |
| $m_2$ | 90.70 | 90.26 | 90.94 | 91.14 |
| $m_3$ | 89.87 | 90.78 | 89.80 | 90.63 |
| $m_4$ | 90.23 | 90.96 | 90.59 | 89.33 |

(a) $m$ denotes the best models, and its subscript denotes the index of training they are from. The average improvement from SoE is 0.85.

| $\pi_a$ \ $\pi_b$ | $m_1$ | $m_2$ | $m_3$ | $m_4$ |
|------|-------|-------|-------|-------|
| $m_1$ | 89.24 | 90.00 | 89.23 | 89.04 |
| $m_2$ | 89.82 | 89.80 | 89.89 | 89.47 |
| $m_3$ | 89.31 | 89.50 | 89.21 | 89.70 |
| $m_4$ | 89.25 | 89.38 | 89.75 | 89.07 |

(b) $m$ denotes the top 4 models from the same training, and its subscript denotes the index. The average improvement from combinations is 0.20.

Table 3: Detailed metric scores of planTF and its SoE version tested in Test14-random CL-NR. $m_*$ denotes the index of model. The model on the left of $+$ is $\pi_a$ and on the right is $\pi_b$. Col :Collision, Dri: Drivable area compliance, MP: Making progress, Dir: Direction compliance, P: Progress, TTC: Time To Collision, S: Speed compliance, Com: Comfort. The (*) signifies the metric is multiplicative, and (number) indicates the weight of additive metric. Green emphasizes the best performance.

| Model | CLN | Col(*) | Dri(*) | MP(*) | Dir(*) | P(5) | TTC(5) | S(4) | Com(2) |
|---|---|---|---|---|---|---|---|---|---|
| $m_1$ | 86.33 | 94.06 | 97.32 | 98.85 | 100 | 91.04 | 88.89 | 96.21 | 97.70 |
| $m_2$ | 84.87 | 94.64 | 95.02 | 99.23 | 99.81 | 88.35 | 91.57 | 97.93 | 93.49 |
| $m_3$ | 86.74 | 95.40 | 95.40 | 100 | 99.81 | 91.50 | 92.34 | 97.25 | 94.64 |
| $m_4$ | 85.73 | 91.95 | 96.93 | 100 | 99.23 | 92.54 | 90.04 | 96.77 | 92.72 |
| $m_1+m_2$ | 85.71 | 93.68 | 96.17 | 99.23 | 100 | 90.54 | 89.27 | 97.30 | 97.70 |
| $\mathbf{m_1+m_3}$ | **88.16** | **95.21** | **96.93** | **99.62** | **100** | **91.69** | **92.72** | **96.69** | **97.32** |
| $m_1+m_4$ | 87.18 | 94.83 | 96.55 | 100 | 99.43 | 91.70 | 90.80 | 96.57 | 98.08 |
| $\mathbf{m_2+m_4}$ | **88.03** | **95.59** | **97.32** | **99.62** | **99.81** | **91.14** | **91.95** | **97.53** | **94.64** |

Table 4: The CL-NR results of Pluto-SoE using Train14(T14) as validation set and Val14(V14) as test set. $T14_{val}$ is the validation score of SoE policy in Train14 set. $V14_{test}$ ($T14_{val}$) is the test score in Val14 set, using Train14 as validation set for selecting SoE policy. $V14_{test}$ ($V14_{val}$) use Val14 for both validation and test, representing the best possible result. $m_n^*$ and $m_n$ are different checkpoints depending on validation set.

(a) The policy improvement $\theta$. Green values are improved. Red values are worsen

| $\pi_a$ \ $\pi_b$ | $T14_{val}$ score | | | | $V14_{test}$ score ($T14_{val}$) | | | | $V14_{test}$ score ($V14_{val}$) | | | |
|---|---|---|---|---|---|---|---|---|---|---|---|---|
| | $m_1$ | $m_2$ | $m_3^*$ | $m_4$ | $m_1$ | $m_2$ | $m_3^*$ | $m_4$ | $m_1$ | $m_2$ | $m_3$ | $m_4$ |
| $m_1$ | 0 | 0.68 | 0.18 | 0.18 | 0 | 0.64 | 0.51 | 0.92 | 0 | 0.64 | 0.03 | 0.92 |
| $m_2$ | 0.54 | 0 | 0.22 | 0.83 | 0.43 | 0 | 0.09 | 0.88 | 0.43 | 0 | 0.67 | 0.88 |
| $m_3^{(*)}$ | 0.26 | 0.33 | 0 | -0.63 | 0.39 | 0.10 | 0 | 0.48 | 0.07 | 0.51 | 0 | 0.83 |
| $m_4$ | 0.12 | 0.72 | -0.76 | 0 | 0.79 | 0.70 | 0.59 | 0 | 0.79 | 0.70 | 0.78 | 0 |

(b) The overall score. The darkness of color is determined by the difference to mean diagonal value.

| $\pi_a$ \ $\pi_b$ | $T14_{val}$ score | | | | $V14_{test}$ score ($T14_{val}$) | | | | $V14_{test}$ score ($V14_{val}$) | | | |
|---|---|---|---|---|---|---|---|---|---|---|---|---|
| | $m_1$ | $m_2$ | $m_3^*$ | $m_4$ | $m_1$ | $m_2$ | $m_3^*$ | $m_4$ | $m_1$ | $m_2$ | $m_3$ | $m_4$ |
| $m_1$ | 89.0 | 90.7 | 89.4 | 90.1 | 89.4 | 90.9 | 90.0 | 90.4 | 89.4 | 90.9 | 89.8 | 90.4 |
| $m_2$ | 90.6 | 90.0 | 90.2 | **90.9** | 90.7 | 90.3 | 90.4 | **91.2** | 90.7 | 90.3 | 90.9 | **91.2** |
| $m_3^{(*)}$ | 89.5 | 90.4 | 89.2 | 89.3 | 89.8 | 90.4 | 88.7 | 89.8 | 89.8 | 90.8 | 89.8 | 90.6 |
| $m_4$ | 90.1 | 90.7 | 89.2 | 90.0 | 90.2 | 91.0 | 89.9 | 89.3 | 90.2 | 91.0 | 90.6 | 89.3 |

## 5.3 EMPIRICAL STUDIES AND KEY FINDINGS

**The Robustness of SoE Selection.** To validate the robustness of the SoE policy selection, we designate the Train14 split as the validation set and the Val14 split as the test set, reporting the CL-NR results of Pluto in Table 4. We observe that the policy improvements shown in Table 4a are almost positive. This indicates that SoE is generally more effective than the expert policies it combines. However, the magnitude of the improvement provided by SoE varies, necessitating the use of the validation set to select the optimal SoE policy. In Table 4b, we observe that the optimal combination selected on Train14 is also the optimal combination on Val14, and the distribution of SoE policy scores is similar across both scenario sets. Therefore, if the validation and test sets share a similar distribution, the SoE approach can successfully select a superior strategy via the validation set.

However, cases exist where the best combination on the validation set is not the best on the test set. This is evident in the results for the CL-R mode as shown in the Appendix D. Even in this scenario, we still observe that the policy improvements are nearly all positive, which continues to demonstrate that the SoE method effectively enhances policy performance. The difference lies in the fact that the best combination found on Train14 differs from the best combination found on Val14, although

the performance gap between the two is small. In summary, the greater the discrepancy between the test and training sets, the more the performance of SoE depends on the similarity between the validation and test sets. In practical application, the distribution of the training set can be optimized to better resemble the test set, which would facilitate the more accurate identification of the optimal SoE strategy.

**Why does SoE work?**     Table 3 shows the detailed metric scores of planTF and its SoE version. For the original policy, the best performance of certain metrics scatters across models $m$ from different training. The bolded SoE version, $m_1 + m_3$ and $m_2 + m_4$, achieving significant performance improvements. Regarding their sub-term scores, we found that the SoE typically combines the performance of components in a manner like linear interpolation. Sometimes, the combination surpasses the maximum score of components, but is still capped by the best possible score that planTF can achieve. We think this data helps us understand why SoE works. The trained IL planners are naturally imperfect, unable to handle the accumulated errors of all types. It's very hard, or even impossible, to obtain a perfect model that excels in all aspects from a single training. SoE works by combining imperfect models in the time domain, forming a perfect model that is difficult to obtain through training. Additional experiment results for understanding SoE are added in Appendix.C.

**Best Hyperparameter Setting**     SoE introduces only two hyperparameters, the period $n$ and the number of training $m$. We use the average improvement $\lambda$ and the maximal policy improvement $\theta$ to quantify the improvement of SoE; the higher, the better. The parameter sweep of $n$ is visualized in Fig.6 of Appendix C. We can find that 2 is the best value for the period $n$ in both the CL-NR and CL-R tests. A higher value lowers the improvement of SoE. In other words, the alternative inference is the best way to combine knowledge from both models. The number of training $m$ increases the training cost linearly and the validation cost quadratically. The study recommends selecting the highest feasible value of $m$ within computational resource constraints, as a higher $m$ yields a higher probability of obtaining a better model.

**Is More Experts Better?**     For the current scheduling function, we investigate whether utilizing more experts yields superior results. In this experiment, we increase the set of experts $\mathcal{N}$ to include three policies, $\{\pi_a, \pi_b, \pi_c\}$. Building upon our experience with two experts, we design the scheduling function to maximize the frequency of expert switching; specifically, the three policies sequentially participate in inference. The resulting performance is presented in Table 7 of Appendix E. The results indicate that increasing the number of experts does not elevate the upper performance bound of the SoE strategy; however, it significantly reduces the variance in performance among different SoE policies. Notably, in the CL-NR mode, even the worst-performing SoE strategy surpasses the best non-SoE policy. Nevertheless, the SoE method has a drawback that loading a greater number of experts proportionally increases GPU memory consumption. Therefore, when utilizing more experts, a trade-off must be managed between the enhanced policy stability and the increased memory footprint.

**Ablation Study of The Selection of Expert Policy**     The key design in formulating SoE policy is to gather the best model from different trainings. In this ablation study, we modify this design to use $m$ best models from the same training to construct the model set $M$. The result is shown in Table 2b. The average improvement $\lambda$ drops significantly in the new setting, from 0.85% to 0.20%.

## 6   CONCLUSION AND FUTURE WORK

This work proposes Sequence of Expert, a plug-and-play method to improve the performance of IL planners. It requires no additional data, model modifications, or computational overhead, making it a new dimension for scaling IL planners. Through extensive evaluation, we show that SoE is universally effective for all IL planners. In particular, the SoE version of a baseline planner achieves state-of-the-art performance in the nuPlan benchmark, showing its practical potential. We believe that SoE will become a valuable tool for researchers and engineers seeking to achieve the ultimate performance of IL planners. This framework may also work in other domains where IL planners are deployed in a closed-loop environment, such as robot manipulation. We leave the exploration to future works. Due to page limits, limitations are discussed in Appendix.F.

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

# A   SUPPORTING EVIDENCE FOR LIMITATIONS OF IL PLANNERS

This section introduces the supporting evidence for limitations of IL planners. We introduce the limitations in the order in the main text.

**OL–CL mismatch.**   Fig.3 shows the validation score of all the intermediate checkpoints of Pluto w/o, tested in all three modes. We can observe that the planners achieving the best OL imitation accuracy are often not those that deliver the best CL performance. In OL mode, the model from epoch 24 achieves the best performance, but it doesn't achieve the best CL performance. While OL metrics improve steadily with training, CL metrics fluctuate significantly; for example, the CL-R performance changes substantially during training.

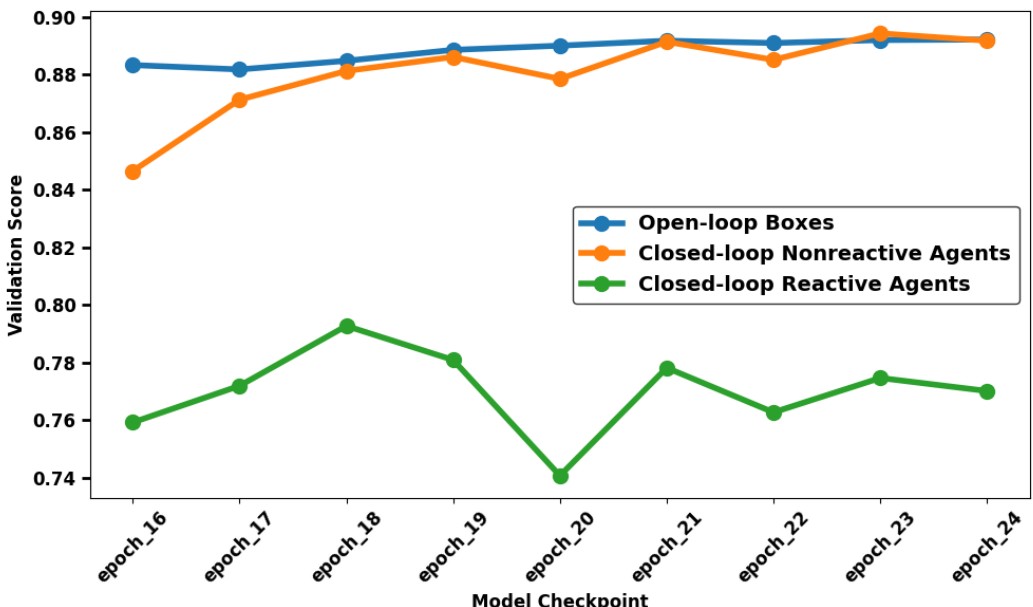

Figure 3: The validation score of all the intermediate checkpoints of Pluto w/o, testing respectively in the open-loop, closed-loop nonreactive, closed-loop reactive modes.

**Early-stage CL superiority.**   As Fig.3 shows, the Pluto w/o achieves best CL-NR performance with the model from epoch 23 and best CL-R performance with the model from epoch 18. This means that the highest CL scores often arise from earlier training epochs rather than from the final converged models. This effect is more pronounced when the deployment distribution diverges strongly from the training distribution, e.g., CL-NR compared to CL-R.

**Inter-run complementarity.**   Models trained with identical architectures and data but different random seeds achieve nearly indistinguishable OL losses, yet their CL performance differs substantially. This limitation is supported by the results in Fig.4, which show the closed-loop reactive validation score for three different training sets. From the figure, we can observe that there is no similarity between their CL-R performance. The best model emerges at different training stages, and its performance changes unpredictably as the training progresses.

However, we find that these models exhibit complementary strengths in handling diverse failure cases. This phenomenon can be observed in Fig.5, which exhibits the validation score of the four best models in $\mathcal{M}$, grouped by the type of scenarios. The key indicator in this figure is the performance difference in the same scenario type. The maximal performance difference is around 2% in the OL test, and it only appears in several scenario types. However, the maximum performance difference can exceed 10% in the CL test and is observed in nearly all scenario types. We hypothesize that the differences in the error accumulation contribute to this phenomenon. This suggests that training variance is not noise but an exploitable source of diversity.

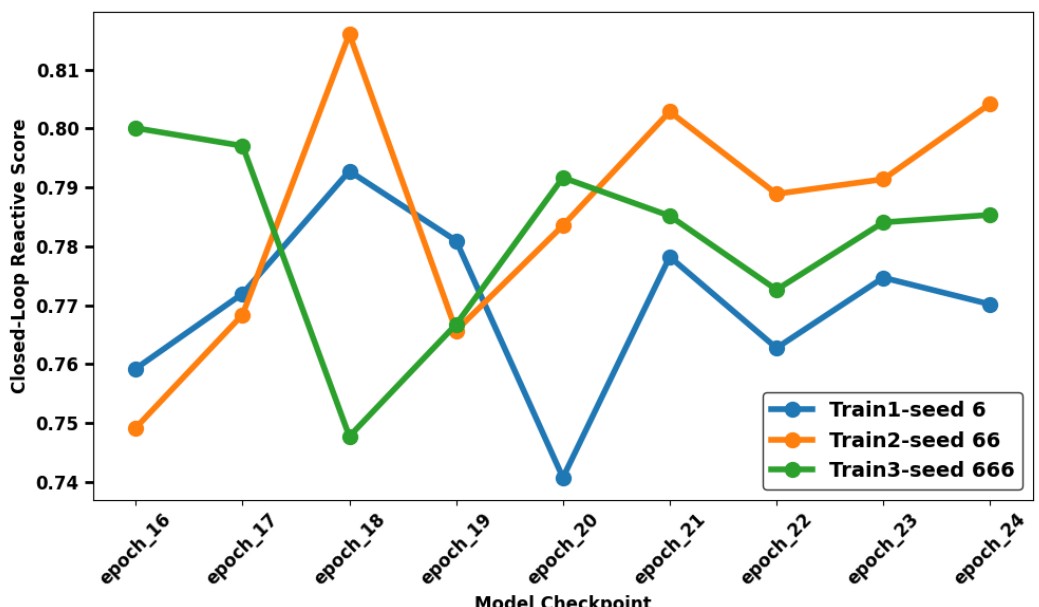

Figure 4: The closed-loop reactive validation score of all the intermediate checkpoints of Pluto w/o. The results are from three different training sessions, in which all training settings remain unchanged except for the random seed.

## B   EVALUATION DETAILS

### B.1   THE SCENARIO SPLITS

- Val14: Val14 contains 1,118 scenarios from 14 scenario types, which is claimed to be an excellent proxy for nuPlan's online benchmark. The details scenario compositions are: low-magnitude-speed(100), near-multiple-vehicles(85), starting-straight-traffic-light-intersection-traversal(98), stopping-with-lead(93), traversing-pickup-dropoff(99), starting-left-turn(100), starting-right-turn(98), high-magnitude-speed(99), changing-lane(70), high-lateral-acceleration(96), stationary-in-traffic(98), behind-long-vehicle(14), waiting-for-pedestrian-to-cross(53), following-lane-with-lead(15).

- Train14: Train14 and Val14 contain the same number and distribution of scenarios, but all test scenarios within Train14 are sourced exclusively from the original training data. This ensures zero scenario overlap between the two sets. The scenarios were not manually selected but were generated in a one-shot process via random number generation.

- Test14-random: Test14-random contains 261 randomly selected scenarios from 14 scenario types. It's data distribution is highly biased from Val14. It's often used for performance comparison in certain literatures.

### B.2   THE MEANING OF INDIVIDUAL METRICS

We detail the meaning of all the metrics in the nuPlan Benchmark. It can be found in `https://github.com/motional/nuplan-devkit`. The (*) sign signifies the metric is multiplicative, and (number) indicates the weight of the additive metric.

- No at-fault Collisions(*): A collision is defined as the event of the ego's bounding box intersecting another agent's bounding box. This metric contributes to the scenario score as a multiplier based on the number of at-fault collisions that happen in a scenario for each group and the acceptable thresholds.

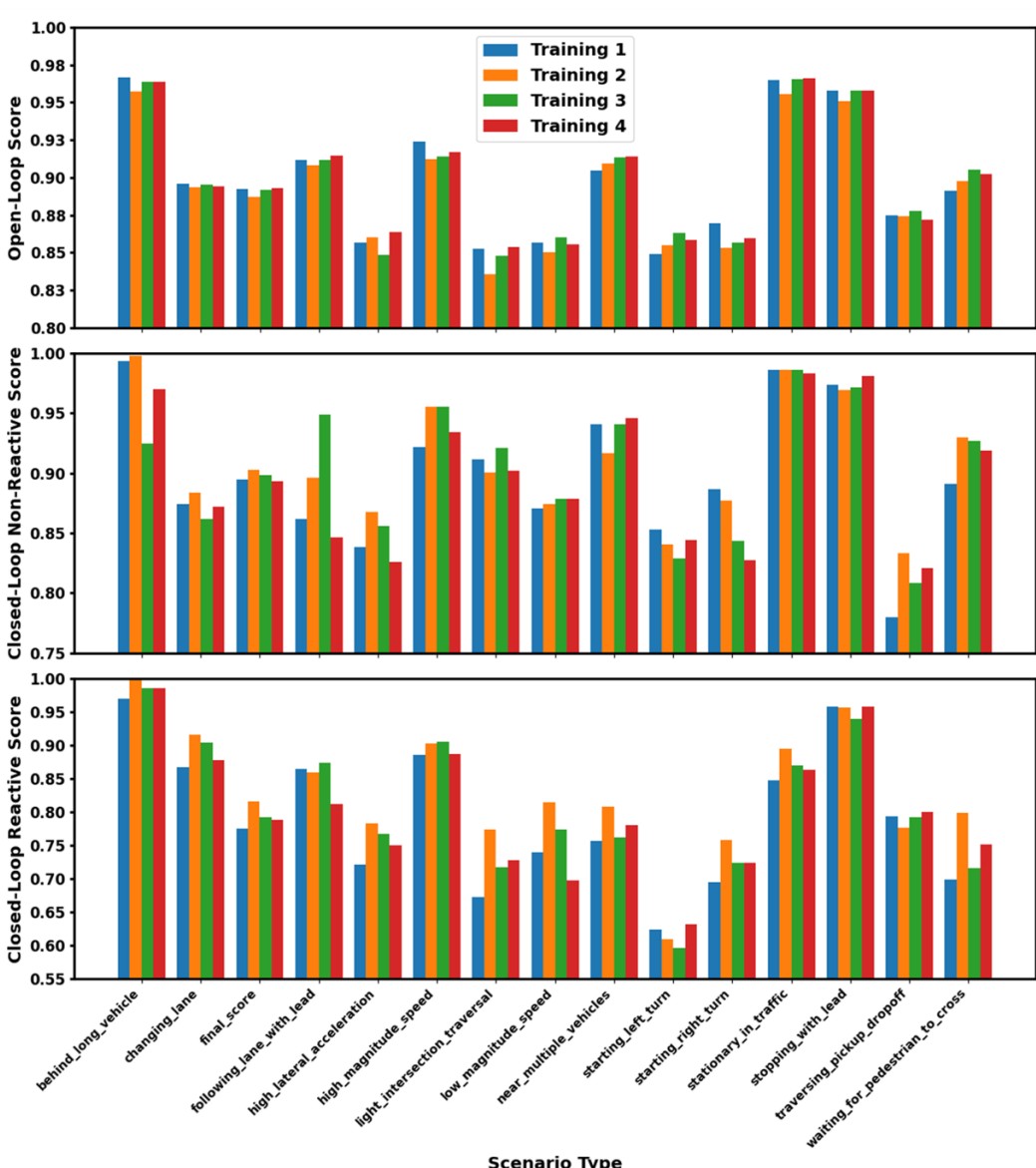

Figure 5: The validation score of 4 best models in $\mathcal{M}$, grouped by the type of scenarios. The models in $\mathcal{M}$ are selected from four different training sessions with various random seeds. The type of scenarios is defined by the nuPlan benchmark. Note that the scale of the y-axis differs a lot.

- Drivable area compliance(*): The ego should drive in the mapped drivable area at all times. The drivable area compliance metric identifies the frames when the ego drives outside the drivable area. This boolean metric contributes to the scenario score as a multiplier.

- Driving direction compliance(*): This is a metric defined to penalize ego when "it drives into oncoming traffic". The score is set to 1 if it does not drive/move against the flow and 0 if it drives against the flow more than 6m.

- Making progress(*): This metric is defined as a Boolean metric based on the "Ego progress along the expert's route ratio". Its score is set to 1 if the ratio exceeds the selected threshold and is set to 0 otherwise.

- Time to Collision (TTC) within bound(5): TTC is defined as the time required for the ego and another track to collide if they continue at their present speed and heading. The Boolean metric contributes to the scenario score in the weighted average function.

- Ego progress along the expert's route ratio(5): It evaluates the progress of the driven ego trajectory in a scenario by comparing its progress along the route that the expert takes in that scenario. The expert's route is extracted as a sequence of lanes and lane connectors that it moves along during the scenario. The ratio contributes to the scenario score in the weighted average function.

- Speed limit compliance(4): This metric evaluates if the ego's speed exceeds the associated speed limit in the map. The speed limit is queried from the lane ego is associated with. This metric contributes to the scenario score in the weighted average function.

- Comfort(2): It measures the comfort of the ego's driven trajectory by evaluating minimum and maximum longitudinal accelerations, maximum absolute value of lateral acceleration, maximum absolute value of yaw rate, maximum absolute value of yaw acceleration, maximum absolute value of longitudinal component of jerk, and maximum magnitude of jerk vector.

### B.3  DETAILS OF TABLE 1

We have the following general rules to derive the result:

- If the baseline provides an open-source model, we use the model it offers to verify the performance. If no open-source model is provided, we select the best result from multiple training runs in SoE framework as its performance.

- During SoE Formulation, we do not make any modifications to the open-source implementation of the baseline if possible.

- For the IL planners generating multiple trajectories, we select the trajectory with the highest probability/confidence.

- The SoE Formulations are all one-shot, which means that all models are trained exactly $m$ times.

- The models and codes for reproducing results in Table 1 are open-sourced.

Following these rules, we enumerate the source of results in Table 1 as follows:

- IDM(Treiber et al., 2000): This planner is rule-based and can be found in `https://github.com/motional/nuplan-devkit`.

- PDM series(Dauner et al., 2023): The open-source link is: `https://github.com/autonomousvision/tuplan_garage`. PDM series doesn't provide open checkpoints, so its performance is reported from the best in $\mathcal{M}$ of SoE.

- Raster model(Caesar et al., 2021): We use the integrated implementation of Raster Model in nuPlan `https://github.com/motional/nuplan-devkit`. It doesn't have open checkpoints, and its performance is reported as the best in $\mathcal{M}$ of SoE.

- PlanTF(Cheng et al., 2024b): It's open-source link `https://github.com/jchengai/planTF`. It provides an open-sourced checkpoint, so we use it to represent its performance. We then formulate SoE with its training code.

- Pluto(Cheng et al., 2024a): It's open-source link `https://github.com/jchengai/pluto`. It provides an open-sourced checkpoint, so we use it to represent its performance. We then formulate SoE with its training code. SoE1 and SoE2 are two different SoE policies. The difference is that SoE1 adopts CL-NR as its validation set in SoE formulation, while SoE2 uses CL-R. It provides a rule-based fallback strategy, which we use without modification.

- DiffusionPlanner(Zheng et al., 2025): It's open-source link `https://github.com/ZhengYinan-AIR/Diffusion-Planner`. It provides an open-sourced checkpoint, so we use it to represent its performance. However, it didn't provide the rules to reproduce the result of their hybrid version. We use the result from their paper directly and cannot measure its inference time. Due to resource limitations, we leave the formulation of its SoE for future work.

## C  ADDITIONAL RESULT ON WHY SoE WORKS

**The parameter sweep of $n$.**  Here we list the parameter sweep of $n$ in Val14 test, including results in both CL-NR and CL-R tests. The validation set for SoE selection is also Val14 split.

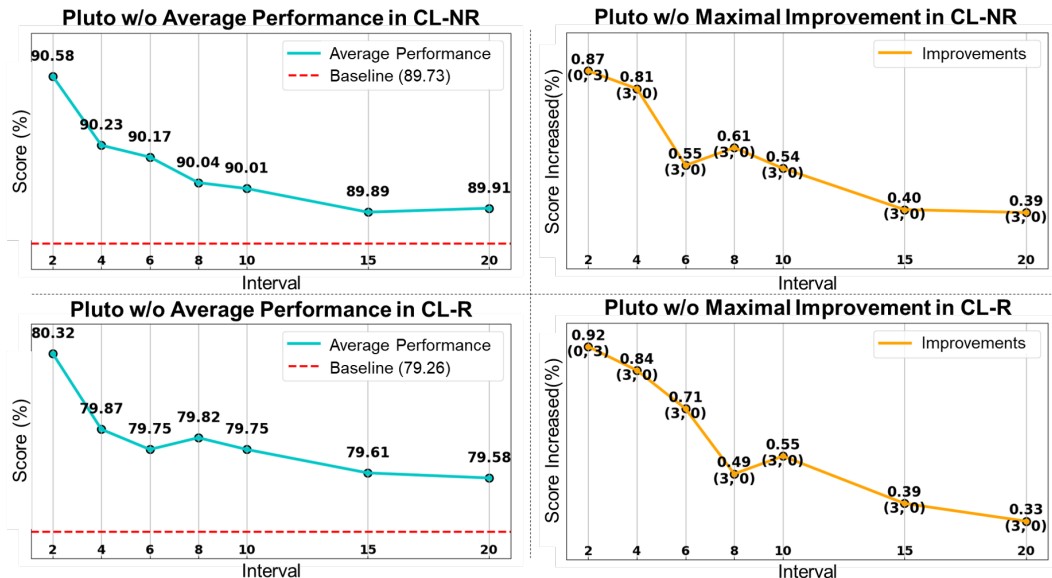

Figure 6: The parameter sweep of $n$ in Val14 test. We use the average improvement $\lambda$ and the maximal policy improvement $\theta$ to quantify the improvement of SoE. The annotation in maximal improvement marks the value and the models' indices achieving maximal improvement.

Table 5: The number of scenarios containing critical failure between baselines and their SoE version, tested in Val14 CL-NR.In its first row, $\pi$ denotes the scenario fails with this policy, while $!\pi$ denotes not. $\&$ is the logical AND, e.g. $\pi_a \& \pi_b$ means the number of scenarios where both policy fail.

| Planner | Error Type | $\pi_a \& \pi_b \to \pi^n$ | $(\pi_a \& !\pi_b, !\pi_a \& \pi_b) \to \pi^n$ | $!\pi_b \& !\pi_b \to \pi^n$ |
|---|---|---|---|---|
| planTF | Collision | $38 \to 36$ | $(46,35) \to 26$ | $0 \to 3$ |
| | Drivable | $18 \to 17$ | $(18,24) \to 9$ | $0 \to 1$ |
| Pluto | Collision | $23 \to 23$ | $(19,20) \to 9$ | $0 \to 4$ |
| | Drivable | $6 \to 6$ | $(5,3) \to 0$ | $0 \to 0$ |

**SoE reduces the scenarios where only one policy fails.** Table 5 provides additional context for understanding SoE, listing how the number of scenarios containing critical failures changes between SoE and its components. It shows that SoE mainly reduces the scenarios where only one policy fails, e.g., from (19, 20) to 9 in collisions of Pluto, meaning that it requires one capable policy to lead another policy. This supports the hypothesis that SoE works by leveraging the difference in ability in erasing accumulated errors.

## D  THE ROBUSTNESS OF SOE SELECTION.

We designate the Train14 split as the validation set and the Val14 split as the test set, reporting the CL-R results of Pluto in Table 6.

## E  RESULTS ABOUT MORE EXPERTS.

Table7 introduce the results that increased expert number from 2 to 3.

Table 6: The CL-R results using Train14(T14) as validation set and Val14(V14) as test set. $T14_{val}$ is the validation score of SoE policy in Train14 set. $V14_{test}$ $(T14_{val})$ is the test score in Val14 set, using Train14 as validation set for selecting SoE policy. $V14_{test}$ $(V14_{val})$ use Val14 for both validation and test, representing the best possible result. $m_n^*$ and $m_n$ are different checkpoints depending on validation set.

(a) The policy improvement $\theta$.

| $\pi_b$ / $\pi_a$ | $T14_{val}$ score | | | | $V14_{test}$ score $(T14_{val})$ | | | | $V14_{test}$ score $(V14_{val})$ | | | |
|---|---|---|---|---|---|---|---|---|---|---|---|---|
| | $m_1$ | $m_2$ | $m_3^*$ | $m_4^*$ | $m_1$ | $m_2$ | $m_3^*$ | $m_4^*$ | $m_1$ | $m_2$ | $m_3$ | $m_4^*$ |
| $m_1$ | 0 | 0.13 | 0.23 | -0.79 | 0 | 0.19 | 0.76 | 0.17 | 0 | 0.19 | 0.68 | -0.59 |
| $m_2$ | 0.05 | 0 | 0.61 | 1.13 | 0.3 | 0 | 0.21 | 0.17 | 0.3 | 0 | 0.59 | 0.28 |
| $m_3^{(*)}$ | 0 | 0.39 | 0 | 1.24 | 0.79 | -0.01 | 0 | 0.54 | 1.01 | 0.76 | 0 | 0.45 |
| $m_4^{(*)}$ | -0.61 | 0.81 | 1.16 | 0 | 0.25 | 0.03 | 0.44 | 0 | -0.41 | 0.06 | 0.32 | 0 |

(b) The overall score. The darkness of color is determined by the difference to mean diagonal value.

| $\pi_b$ / $\pi_a$ | $T14_{val}$ score | | | | $V14_{test}$ score $(T14_{val})$ | | | | $V14_{test}$ score $(V14_{val})$ | | | |
|---|---|---|---|---|---|---|---|---|---|---|---|---|
| | $m_1$ | $m_2$ | $m_3^*$ | $m_4^*$ | $m_1$ | $m_2$ | $m_3^*$ | $m_4^*$ | $m_1$ | $m_2$ | $m_3^*$ | $m_4^*$ |
| $m_1$ | 77.2 | 80.0 | 78.6 | 78.0 | 79.3 | 81.8 | 80.0 | 79.4 | 79.3 | 81.8 | 80.7 | 79.1 |
| $m_2$ | 80.0 | 79.9 | 80.5 | **81.0** | **81.9** | 81.6 | 81.8 | 81.8 | 81.9 | 81.6 | 82.2 | 81.9 |
| $m_3^{(*)}$ | 78.4 | 80.3 | 78.4 | 80.1 | 80.1 | 81.6 | 79.2 | 79.7 | 81.0 | **82.4** | 80.0 | 80.5 |
| $m_4^{(*)}$ | 78.2 | 80.7 | 80.0 | 78.8 | 79.5 | 81.6 | 79.6 | 78.8 | 79.3 | 81.7 | 80.3 | 79.7 |

# F  LIMITATIONS

SoE also exhibits several limitations. Although SoE doesn't require additional computation in deployment, it introduces more overhead to training and evaluation. There may be other approaches to derive complementary models in addition to complete training. We leave this for future work. SoE leverages model variance to significantly improve model performance. However, the possible top performance is still determined by other dimensions, such as model and data. SoE works as an additional dimension to improve IL planners, not the solution for full autonomous driving.

# G  THE USE OF LARGE LANGUAGE MODELS

This paper utilizes an LLM solely for language polishing, enhancing the quality of writing expression.

Table 7: Ablation study on more experts. The CL-NR score reference is $m_0 : 89.44$, $m_1 : 90.27$, $m_2 : 89.80$, $m_3 : 89.33$. The CL-NR score reference is $m_0 : 79.28$, $m_1 : 81.60$, $m_2 : 80.01$, $m_3 : 79.70$

| Absent model Index | Model Index Sequence | CL-NR Score | CL-R Score |
|---|---|---|---|
| 0 | (1, 2, 3) | 91.22 | 81.73 |
|   | (1, 3, 2) | 91.27 | 81.57 |
|   | (2, 1, 3) | 91.11 | 81.95 |
|   | (2, 3, 1) | 91.23 | 81.87 |
|   | (3, 1, 2) | 91.20 | 81.79 |
|   | (3, 2, 1) | 91.21 | 81.80 |
| 1 | (0, 2, 3) | 90.25 | 80.42 |
|   | (0, 3, 2) | 90.51 | 80.10 |
|   | (2, 0, 3) | 90.45 | 80.43 |
|   | (2, 3, 0) | 90.48 | 80.45 |
|   | (3, 0, 2) | 90.41 | 80.40 |
|   | (3, 2, 0) | 90.19 | 80.25 |
| 2 | (0, 1, 3) | 90.92 | 81.73 |
|   | (0, 3, 1) | 90.68 | 81.38 |
|   | (1, 0, 3) | 90.85 | 81.68 |
|   | (1, 3, 0) | 90.88 | 81.48 |
|   | (3, 0, 1) | 90.63 | 81.33 |
|   | (3, 1, 0) | 90.73 | 81.45 |
| 3 | (0, 1, 2) | 91.17 | 81.91 |
|   | (0, 2, 1) | 90.97 | 82.19 |
|   | (1, 0, 2) | 91.16 | 81.98 |
|   | (1, 2, 0) | 90.92 | 82.35 |
|   | (2, 0, 1) | 91.22 | 81.93 |
|   | (2, 1, 0) | 91.06 | 82.31 |

