# OpenReview forum: "Sequence of Expert: Boosting Imitation Planners for Autonomous Driving through Temporal Alternation"
_ICLR.cc/2026/Conference — Submitted to ICLR 2026_

### Official Review · Reviewer_4i59 · 2025-10-23

**Soundness:** 3
**Presentation:** 3
**Contribution:** 2
**Rating:** 4
**Confidence:** 5

**Summary:**

The paper "Sequence of Expert: Boosting Imitation Planners for Autonomous Driving Through Temporal Alternation" proposes a new method termed "sequence of Experts (SoE) to enhance the pipeline's closed-loop performance without modifying the model architecture or applying data augmentation. The idea is to train multiple runs of the same planner architecture with different random seeds and, at inference time, alternate between the original policy and these "expert" models at fixed temporal intervals.

**Strengths:**

- The paper addresses an important problem in autonomous driving, that is, the performance gap between open-loop training and closed-loop evaluation.
-  The proposed approach is conceptually simple and easy to integrate into existing systems without retraining from scratch.
- The authors provide empirical evaluations on the nuPlan benchmark and demonstrate improvements over baseline imitation planners.

**Weaknesses:**

- Unclear scheduling mechanism:

The description of the temporal scheduling mechanism (Section 4.2) is insufficiently detailed. It is unclear whether the scheduling function $\delta$ is a predefined periodic function or a trainable component. Additionally, the relationship between the policy $\pi^*$ and the final SoE policy should be explicitly clarified.

- Limited practicality for deployment:

Although the method is described as plug-and-play, its deployment feasibility is questionable. Maintaining multiple large expert models (e.g., >1 GB each for UniAD-style planners) poses serious storage and memory challenges for real-world on-board systems.

- Lack of theoretical grounding:

The approach relies on the empirical assumption that training with different random seeds yields complementary closed-loop behaviors. This assumption lacks theoretical justification and may not generalize across tasks or architectures.

**Questions:**

1. What exactly is the scheduling function $\delta$?
2. In the section 5.1, $f()$ denotes the nuPlan validation score of a policy, is it the same as the notation in Eq. (4)?
3. What is the meaning of "R", "L", and "H" in Table I?

---

> ### Author Response · Authors · 2025-12-03
> **Rebuttal by Authors**
>
> We found rebuttal button missing and use official comment for rebuttal. We thank Reviewer 4i59 for the helpful feedback, questions, and suggestions.
>
> ## Weakness 1
>
> Thank you for your suggestion. We have rewritten Section 4.2 to make the description clearer. The scheduling function is a class of functions that select the model to be used for inference based on the current simulation time step; the input is the time step, and the output is the index of the selected model. The "Temporal Scheduling" paragraph explicitly defines the scheduling function used in this paper, which is a predefined periodic function. We have also revised the notation for $\pi^*$ and $\pi^o$ to $\pi_a$ and $\pi_b$ to express the equal standing of the two policies. Furthermore, we have used Eq. (4) to describe their relationship with the SoE policy more clearly.
>
> **Relevant modifications to paper:** Improving the contents of Section 4.2.
>
> ## Weakness 2
>
> Thank you for your suggestion. SoE indeed increases VRAM consumption proportionally, but we believe this cost is acceptable at the current stage. Current end-to-end autonomous driving models in the industry are not exceptionally large. Taking the UniAD model you mentioned as an example, assuming the model's inference precision is bfloat16, it would consume approximately 3GB of VRAM. Mainstream on-board Systems-on-Chip, such as the Jetson Orin NX, currently feature 16GB of shared memory. Furthermore, these models do not occupy excessive CPU memory during inference, which leads us to believe that SoE holds certain practicality for deployment.
>
> At the same time, we believe that real-time inference models will not become substantially larger in the future because inference latency is a bottleneck for autonomous driving models. Since inference latency is typically positively correlated with the number of parameters, models with an extremely large parameter count may fail to meet real-time requirements. When the model size can no longer be increased, SoE presents a promising dimension to continue scaling model performance.
> Additionally, we are also exploring methods for multiple expert models to share a backbone network, a technique that has the potential to significantly reduce the VRAM consumption of the SoE approach.
>
> ## Weakness 3
>
> Thank you for your suggestion. Currently, we are unable to provide a strict mathematical proof to explain why these expert models are "complementary." However, we have found related research [1,2] that reports the same phenomenon we observed: that different random seeds lead to qualitatively different behavior under distribution shift.
>
> This variability stems from the non-convex loss surface inherent in neural network training. Neural networks are optimized via stochastic, gradient-based methods, and consequently, different random initializations, data order, and other factors can steer the training process toward distinct local minima in the weight space.
> While these minima may perform similarly on the training data, they often diverge in their generalization capabilities, particularly when faced with an out-of-distribution shift. Researchers in [1] claimed that such variance between independent runs is "unavoidable," even with identical hyperparameters, a finding that is consistent with our own observations.
>
> **Relevant modifications to paper:** Adding an explanation in Paragraph "Expert Selection" of Section 4.2.
>
> ## Question 1
>
> Thank you for your question. The scheduling function is a class of functions that selects the model to be used for inference based on the current simulation time step; the input is the time step, and the output is the index of the selected model. Taking the scheduling strategy shown in Eq. (4) in the paper as an example (which is perhaps the simplest scheduling function), assuming $n=2$, this strategy will alternate between selecting the policies for inference. We have revised Section 4.2 to make its description clearer.
>
> **Relevant modifications to paper:** Improving the contents of Section 4.2..
>
> ## Question 2
> Thank you for your suggestion. There was indeed an issue with notation duplication here. We have revised Eq. (4) to make the expression in that section clearer. There is now no longer any duplicated notation.
>
> ## Question 3
> Thank you for your question. These three letters are abbreviations for the planner classifications due to space constraints in the table. They stand for Rule-based, Learning-based, and Hybrid, respectively. We have added this information to the table caption.
>
> ## Reference
>
> [1]. Keller Jordan. On the variance of neural network training with respect to test sets and distributions. In The Twelfth International Conference on Learning Representations, 2024.
>
> [2]. Pranava Swaroop Madhyastha and Rishabh Jain. On model stability as a function of random seed. In Proceedings of the 23rd Conference on Computational Natural Language Learning (CoNLL), pp. 929–939, 2019.

---

### Official Review · Reviewer_xYof · 2025-10-26

**Soundness:** 2
**Presentation:** 3
**Contribution:** 1
**Rating:** 2
**Confidence:** 3

**Summary:**

This paper addresses the persistent closed-loop performance degradation in imitation-learning (IL) planners, driven by the accumulation of small prediction errors under distribution shift. The proposed Sequence of Experts (SoE) method alternates between independently trained models (using different random seeds) across time steps, aiming to exploit their complementary error characteristics without adding parameters or inference cost. The authors provide empirical observations motivating this variance-based approach and evaluate SoE on the nuPlan benchmark, reporting consistent improvements across multiple planner architectures, including state-of-the-art results for a strong baseline system.

**Strengths:**

S1. Explores temporal alternation rather than architectural scaling to mitigate closed-loop error accumulation, representing a rarely addressed improvement dimension for IL planners.

S2. Introduces zero additional inference cost and requires no model or data modifications, making deployment highly practical.

S3. Demonstrates consistent and meaningful closed-loop performance gains across diverse planners, including achieving SOTA on nuPlan.

S4. Provides empirical evidence on OL–CL mismatch and seed-induced complementarity, offering useful insights into why IL planners degrade in closed-loop settings.

**Weaknesses:**

W1. The claim that different seeds provide complementary error-accumulation behaviors is supported only by empirical observations; the paper lacks a deeper theoretical explanation or dynamic modeling of why such complementarity should reliably occur.

W2. The experts differ solely by random seeds under identical architectures and data, raising concerns about whether this restricted diversity is consistently strong and generalizable beyond the evaluated cases, especially in larger amount of data.

W3. Selecting experts using Val14 and then reporting gains largely based on Val14 introduces a risk of data leakage or overfitting to the validation set, potentially inflating the reported improvements.

W4. While inference cost is unchanged, SoE requires multiple full training runs and quadratic validation combinations, making the total computational cost high and less scalable for large industrial models.

W5. Although the paper claims “additional computational overhead,” the method requires multiple full training runs and maintaining several model instances, which substantially increases both computation and deployment overhead. These costs may exceed those of training a single stronger model, so the computational advantage is not guaranteed to scale. The authors should clarify that the zero-cost claim applies only to inference.

**Questions:**

Same as Weakness.

Other questions:

Q1. Could the authors provide results where expert selection relies **only on the training set** (e.g., training loss or closed-loop proxy) while **evaluation fully on held-out test split**? This would clarify whether the current gains depend on information from the validation distribution.

Q2. Does SoE introduce new types of closed-loop failures—e.g., oscillatory heading changes or unstable lateral control—caused by switching between policies with different behavioral biases?

Q3. Table 1 shows identical inference latency for SoE and the original model. How is policy switching performed without introducing any runtime overhead in memory usage or scheduling?

---

> ### Author Response · Authors · 2025-12-03
> **Rebuttal by Authors**
>
> We found rebuttal button missing and use official comment for rebuttal. We thank Reviewer xYof for the helpful feedback, questions, and suggestions.
>
> ## Weakness 1
> Thank you for your suggestion. Currently, we are unable to provide a strict mathematical proof to explain why these expert models are "complementary." However, we have found related research [1,2] that reports the same phenomenon we observed: that different random seeds lead to qualitatively different behavior under distribution shift.
>
> This variability stems from the non-convex loss surface inherent in neural network training. Neural networks are optimized via stochastic, gradient-based methods, and consequently, different random initializations, data order, and other factors can steer the training process toward distinct local minima in the weight space.
> While these minima may perform similarly on the training data, they often diverge in their generalization capabilities, particularly when faced with an out-of-distribution shift. Researchers in [1] claimed that such variance between independent runs is "unavoidable," even with identical hyperparameters, a finding that is consistent with our own observations.
>
> **Relevant modifications to paper:** Adding an explanation in Paragraph "Expert Selection" of Section 4.2.
>
> ## Weakness 2
> Thank you for your suggestion. Currently, we cannot provide a strict proof for this, but based on our observations and the conclusions of other researchers [1], such variance between independent runs is usually "unavoidable" and can be consistently reproduced as training stabilizes. In our paper, we trained four different model architectures, and each training run was performed one-shot. We were able to observe this phenomenon in all of our training experiments.
>
> **Relevant modifications to paper:** Adding an explanation in Paragraph "Expert Selection" of Section 4.2.
>
> ## Weakness 3
>
> Thank you for your valuable suggestion regarding the robustness of our offline expert pair selection.
> We want to first clarify the relevant background concerning the nuPlan dataset used in this study.
> The canonical usage dictates using Val-14 as the validation set and the nuPlan Online Test as the true test set. However, as the Online Test is currently unavailable, and given that prior work [3] suggested Val-14 provides scores similar to the online results, it has become common practice in the community to use Val-14 for both model selection (validation) and final reporting (testing).
> We acknowledge this introduces a potential data leakage concern.
> To rigorously address this issue, we introduced the Train-14 scenario set.
> It has an identical number and type of scenarios as Val-14.
> Crucially, its testing scenarios are drawn exclusively from the training dataset, ensuring zero scenario overlap with Val-14.
> Furthermore, these scenarios were generated randomly using a one-shot process, eliminating manual selection bias.We then performed our selection and testing using the standard split: Train-14 as the Validation Set and Val-14 as the Test Set.
>
> Analysis of Selection Robustness (Table 4):
>
> - General Performance (Table 4a):
> We observe that the performance improvement provided by SoE is overwhelmingly positive, meaning that SoE is almost always stronger than its constituent expert policies. However, the magnitude of the improvement varies, necessitating the use of a validation set to select the optimal SoE strategy.
>
> - Robustness of Optimal Selection (Table 4b):
>     1. In Table 4b, we use color-coding to visualize performance relative to the average score (deeper color indicates greater improvement or degradation).
>     2. We observe that the optimal pair selected on the Train-14 (Validation) set remains the optimal pair on the Val-14 (Test) set.
>     3. Furthermore, the distribution of scores achieved by SoE across the two scenario sets is highly similar.
>
> - Conclusion on Robustness: Our results demonstrate that if the validation and test distributions are generally similar, the SoE strategy chosen via the validation set is indeed robust and likely to be the best performing on the unseen test set.
>
> **Relevant modifications to paper:** Adding New Paragraph "The Robustness of SoE Selection" in Section 5.3.

---

> ### Author Response · Authors · 2025-12-03
> **Additional Rebuttal by Authors**
>
> ## Weakness 4
>
> Thank you for your suggestion. The need to train multiple models is indeed one of the current drawbacks of SoE (Selection of Experts). Beyond seed-induced model variance, we plan to explore other, lighter-weight methods for obtaining diversified models, such as supervised fine-tuning, in future work.
>
> Despite this drawback, we believe SoE still has practical value from two perspectives. First, SoE offers a solution where multiple models with smaller training requirements can replace a single, larger model. Taking the Pluto and Diffusion Planner models in the paper as examples, both models were trained using $1M$ scenarios. Pluto has approximately $4M$ parameters and was trained for 25 epochs, while the Diffusion Planner has $6M$ parameters and was trained for 500 epochs. This results in the training of the two models having a computational difference of roughly $(6/4) \times (500/25) = 30$ times (Pluto is $4M$, 25 epochs; Diffusion Planner is $6M$, 500 epochs. Relative training cost: $(6/4) \times (500/25) = 1.5 \times 20 = 30$). Even if SoE trains the Pluto model four times, its computational requirement is still far less than training the Diffusion Planner once.From the second perspective, we believe that the factors currently hindering the deployment of autonomous driving technology are still related to model performance rather than the cost of model training. Furthermore, current autonomous driving models in the industry are not exceptionally large (e.g., UniAD has only $0.13B$ parameters), and training a model multiple times does not impose a significant cost burden. Therefore, we believe that SoE is a technique worth exploring and using, even in an industrial setting.
>
> ## Weakness 5
>
> Thank you for your suggestion. We have revised the paper to clarify that all mentions of computational overhead refer to the test-time computational overhead.
>
> ## Question 1
>
> Please refer to Weakness 3.
>
> ## Question 2
>
> Thank you for your question. Regarding your examples, SoE does not cause issues related to comfort. Oscillatory heading changes or unstable lateral control are captured in the nuPlan comfortable metric. As shown in the last column of Table 3, SoE does not degrade the comfort metrics and, in fact, improves the comfort scores in many cases.
>
> As for other aspects, such as collisions, SoE indeed has the potential to introduce new failures. However, the number of failures introduced is far less than the number of problems it solves, so overall, SoE still improves model performance. Table 5 in the Appendix details this result. For driving area issues, SoE eliminates all problems and introduces no new ones. For collision issues, SoE reduces collisions in 10 scenarios but increases collisions in 4 scenarios.
>
> ## Question 3
>
> Thank you for your question. The current model scheduling strategy can be implemented with just a single if statement, which requires only a few clock cycles at the CPU level to execute. Consequently, it introduces almost no runtime computational overhead.
> However, SoE does indeed increase VRAM consumption proportionally. We have added a discussion in the experiments section to clearly state that SoE does not increase test-time computational overhead but will proportionally increase VRAM usage.
>
> **Relevant modifications to paper:** Discussion about memory cost in "Is More Experts Better" of Section 5.3.
>
> ## Reference
> [1]. Keller Jordan. On the variance of neural network training with respect to test sets and distributions.
> In The Twelfth International Conference on Learning Representations, 2024.
>
> [2]. Pranava Swaroop Madhyastha and Rishabh Jain. On model stability as a function of random seed. In
> Proceedings of the 23rd Conference on Computational Natural Language Learning (CoNLL), pp.
> 929–939, 2019.
>
> [3]. Daniel Dauner, Marcel Hallgarten, Andreas Geiger, and Kashyap Chitta. Parting with misconceptions
> about learning-based vehicle motion planning. In Conference on Robot Learning, pp. 1268–1281.
> PMLR, 2023.

---

### Official Review · Reviewer_gkWn · 2025-10-29

**Soundness:** 3
**Presentation:** 3
**Contribution:** 2
**Rating:** 2
**Confidence:** 5

**Summary:**

This research addresses a critical limitation of imitation learning in autonomous driving, where small errors accumulate over time leading to failures in closed-loop scenarios. The proposed "Sequence of Experts" (SoE) method improves performance by alternating between different trained models at specific intervals, leveraging their diverse error profiles for enhanced robustness. As a simple and adaptable solution, SoE significantly boosts the performance of existing imitation learning systems without requiring architectural changes or additional data.

**Strengths:**

* A very simple approach: just alternate experts (SoE) every 2nd timestamp
* Working on nuPlan, a quite widely used benchmark

**Weaknesses:**

* Obvious weakness: validation set should be VERY close in terms of distribution to the test set in order to find the correct combination of experts for SoE
* No any ablations / exploration on whether exists a situation when the best combination of experts on val is not the best on the test
* Straightforward drawback: need to wait (and spoil resources) for training multiple models in order to include them into SoE (and usage of different ckpts during one training cycle is not the best strategy according to Table 2)
* Diffusion-based planner is still the best according to the Table 1

Overall, the approach is SUPER simple and no any theoretic considerations: let's just train with multiple random seed different planners and then alternate between them every n'th timestamp.

**Questions:**

* Is the scheduling function $\sigma(t)$ defined in Eq. (4)?

---

> ### Author Response · Authors · 2025-12-03
> **Rebuttal by Authors**
>
> We found rebuttal button missing and use official comment for rebuttal. We thank Reviewer gkWn for the helpful feedback, questions, and suggestions.
>
> ## Weakness 1&2
> Thank you for your suggestion. Weakness 1 and Weakness 2 are related, so we will address them together. To verify the impact of the validation set distribution on the selection of the SoE policy and to avoid the data leakage issue in the current paper, we designed the Train14 scenario set. It has the same number of scenarios and scenario distribution as the Val14 scenario set, but its test scenarios are all from the training set, meaning the two sets have no scenario overlap. Furthermore, we did not manually select the scenarios; they were generated via a one-shot random process. We used Train14 as the validation set and Val14 as the test set, reporting the CL-NR results as shown in Table 4.
>
> First, we can observe that the strategy improvements in Table 4a are almost all positive, which means that the SoE strategy is generally stronger than the expert policies that compose it. However, the magnitude of the improvement brought by SoE varies, thus necessitating the use of a validation set to select a better SoE strategy. In Table 4b, we use color fill to represent the increase in score relative to the average score, with deeper colors indicating greater improvement or decrease. It can be observed that the optimal combination we selected on Train14 is also the optimal combination on Val14, and the score distributions of SoE performance are similar across the two scenario sets. Therefore, if the validation set and test set distributions are roughly similar, the validation set can be used to select a good SoE strategy.
>
> However, as you pointed out, there are situations where the best combination on the validation set is not the best on the test set. This occurs with the CL-R results when using Train14 as the validation set. In this case, we can still observe that the strategy improvements are almost all positive, which still proves that the SoE method can effectively enhance policy performance. Yet, the best combination under Train14 achieved a score of 81.8, while the theoretical best score was 82.4.
> In summary, the greater the difference between the test set and the training set, the more critical the dependency of SoE on the similarity between the validation set and the test set becomes. We emphasized this point in the paper.
> This also provides an alternative line of thought. In practical use, we can optimize the distribution of the training set to make it more closely resemble the test set, which can help in more accurately finding the optimal SoE strategy.
>
> **Relevant modifications to paper:** Adding New Paragraph "The Robustness of SoE Selection" in Section 5.3 and Appendix D.
>
> ## Weakness 3
> Thank you for your suggestion. The need to train multiple models is indeed one of the current drawbacks of SoE (Selection of Experts). Beyond seed-induced model variance, we plan to explore other, lighter-weight methods for obtaining diversified models, such as supervised fine-tuning, in future work.
>
> Despite this drawback, we believe SoE still has practical value from two perspectives. First, SoE offers a solution where multiple models with smaller training requirements can replace a single, larger model. Taking the Pluto and Diffusion Planner models in the paper as examples, both models were trained using $1M$ scenarios. Pluto has approximately $4M$ parameters and was trained for 25 epochs, while the Diffusion Planner has $6M$ parameters and was trained for 500 epochs. This results in the training of the two models having a computational difference of roughly $(6/4) \times (500/25) = 30$ times (Pluto is $4M$, 25 epochs; Diffusion Planner is $6M$, 500 epochs. Relative training cost: $(6/4) \times (500/25) = 1.5 \times 20 = 30$). Even if SoE trains the Pluto model four times, its computational requirement is still far less than training the Diffusion Planner once.From the second perspective, we believe that the factors currently hindering the deployment of autonomous driving technology are still related to model performance rather than the cost of model training. Furthermore, current autonomous driving models in the industry are not exceptionally large (e.g., UniAD has only $0.13B$ parameters), and training a model multiple times does not impose a significant cost burden. Therefore, we believe that SoE is a technique worth exploring and using, even in an industrial setting.

---

> ### Author Response · Authors · 2025-12-03
> **Additional Rebuttal by Authors**
>
> ## Weakness 4
>
> You are correct. The SoE version of Pluto still scores lower than the Diffusion Planner in CL-R mode. However, our main intention with Table 1 was to show, through comparison with the Diffusion Planner, that the SoE method can elevate a weaker model to a state-of-the-art level. The primary comparison targets in that table are the non-SoE versions of the planners. Furthermore, the SoE method could be directly applied to the Diffusion Planner and might further enhance its performance. Unfortunately, due to hardware constraints (the Diffusion Planner requires a large amount of GPU memory), we were unable to perform this experiment. We have designated this as one of our future work directions.
>
> ## Question 1
> Thank you for your suggestion, which is very helpful for improving the quality of our paper. Our original intention was to convey that the method proposed in the "Temporal Scheduling" paragraph is an instance of $\sigma(t)$, which is selecting the corresponding model index based on the simulation time step. We found that the writing in this section was somewhat confusing, so we have revised it to express $\sigma(t)$ clearly.
>
> **Relevant modifications to paper:** Improving the contents of Section 4.2.

---

### Official Review · Reviewer_3YTF · 2025-10-31

**Soundness:** 2
**Presentation:** 2
**Contribution:** 2
**Rating:** 2
**Confidence:** 4

**Summary:**

This paper introduces "Sequence of Experts" (SoE), a novel and simple plug-and-play framework designed to mitigate the issue of error accumulation in imitation learning-based autonomous driving planners. Instead of relying on a single, imperfect model, SoE employs a temporal alternation strategy, periodically switching between a primary policy and a pre-selected "expert" policy during closed-loop deployment. The authors hypothesize that this periodic switching disrupts the compounding of errors by introducing corrective actions from a model with a different failure mode, thereby enhancing robustness. The method requires no architectural modifications and no additional training data.

**Strengths:**

1. The paper addresses error accumulation problem from a new perspective. Instead of focusing on improving a single model's architecture or data, it reframes the problem as one of optimal policy deployment. The key insight is that models from different training stages exhibit complementary weaknesses.

2. The authors demonstrate the effectiveness of SoE across a diverse set of baseline planners (rule-based, MLP-based, Transformer-based), proving its broad applicability.

3. The "plug-and-play" nature and simple implementation make it highly practical and immediately applicable for researchers and practitioners in the field.

**Weaknesses:**

1. It seems just blindly switching models over time. The system does not appear to detect or anticipate error accumulation before switching. It switches policies regardless of whether the current policy is performing well or poorly. This could be suboptimal, as it might unnecessarily interrupt a perfectly good trajectory or fail to switch at the most critical moment. Have the authors considered a more intelligent, state-dependent switching strategy (e.g., based on model uncertainty, trajectory deviation, or a learned gating function) that could trigger the expert policy "on-demand"? Such a comparison would better isolate the benefits of switching itself versus the specific temporal strategy proposed.

2. The temporal scheduling, switching models with fixed time interval, doesn't make sense to me. There is ambiguity in the "Expert" Role and Selection Process. The term "Sequence of Experts" implies that the secondary policy (pi^\*) acts as an "expert" in situations where the primary policy (pi^0) fails. However, the paper's description of this relationship lacks quantitative evidence and a clear mechanism. How is the "expertise" of a policy quantified? For example, at the moment of switching, is there any evidence that the chosen "expert" policy (pi^\*) indeed has a higher probability of success or a better understanding of the current state than (pi^0)? The framework would be more convincing if it included a mechanism to justify why (pi^\*) is the "expert" at that specific moment. Without this, the method appears to be more of a "policy alternation" or "policy shuffling" rather than a true expert consultation.

3. The method currently selects only one expert policy (pi^\*) to pair with the primary policy (pi^0). Given that different models might excel in different scenarios or under different evaluation metrics (e.g., one is better at collision avoidance, another at comfort), why limit the pool to a single expert? Have the authors explored a dynamic approach where the system could choose from a larger pool of candidate experts based on the current driving context? The current offline selection of a single, fixed expert seems to underutilize the full potential of model complementarity.

4. The paper states that the method has a low computational overhead because it only runs one model per step. However, deploying SoE requires loading two models into memory (e.g., VRAM). For large-scale models, this could be a non-trivial memory cost. It would be beneficial for the authors to provide a clear analysis of the memory footprint and actual inference latency compared to a single-model baseline.

5. The selection process is performed offline on a validation set. How robust is this selection? For instance, does the best pair (m-i, m-j) on the validation set consistently perform as the best pair on the test set?

**Questions:**

See Weaknesses.

---

> ### Author Response · Authors · 2025-12-03
> **Rebuttal by Authors**
>
> We found rebuttal button missing and use official comment for rebuttal. We thank Reviewer 3YTF for the helpful feedback, questions, and suggestions.
>
> ## Weakness 1
> Thank you for your suggestion. The current SoE mechanism does indeed implement periodic policy switching rather than detecting error accumulation to trigger a switch.
> Our hypothesis for why this simple scheduling improves the driving policy is based on the nature of closed-loop inference: error is constantly accumulating. The frequent, periodic policy switching acts to interrupt this error accumulation, thereby preventing catastrophic policy failure. Thus, this seemingly "blind" switching strategy can still effectively enhance the performance of the Imitation Learning (IL) planner.
> Furthermore, by selecting policies that are similar, the system is more likely to maintain a good trajectory rather than interrupt it. The high switching frequency also ensures that both expert models are given the opportunity to handle critical moments.
> We agree that a state-dependent switching strategy is a highly promising and valuable research direction and leave it to future works.
>
> ## Weakness 2
> Thank you for your advice. We agree with your assessment that the current mechanism is more accurately described as policy alternation rather than a true expert consultation.
> The terminology in the original manuscript has led to some ambiguity, and we have revised the paper to clarify this. We have changed the notation from the primary policy ($\pi^0$) and secondary policy ($\pi^*$) to Expert Policy A ($\pi_a$) and Expert Policy B ($\pi_b$) to emphasize the equality and independent capabilities of the two policies.
> In Figure 5 of Appendix, we attempt to quantify the performance of each policy based on its score across different scenarios. This data reveals that no single policy is strictly superior across all scenarios. Each policy performs well in some situations and poorly in others. This heterogeneity is the core reason we treat them as equals and seek to combine their complementary abilities through switching.
>
> At this stage of our research, we cannot accurately determine whether one policy is strictly better than the other at any given state. However, by employing similar models within the SoE framework, the policy that is switched to is unlikely to drastically reduce the current trajectory's success rate.
> We consider the development of a mechanism to accurately predict the success probability of a policy at a given state—and designing a switching strategy based on that prediction—as a significant and promising direction for future work (which also addresses the concerns you raised in Weakness1).
>
> **Relevant modifications to paper:** Improving the contents of Section 4.2.
>
> ## Weakness 3
> Thank you for your suggestion.
> We have conducted additional experiments where the switching strategy rotates among a pool of three expert policies. The results indicate that while this approach does not necessarily raise the upper bound of the SoE strategy, it does significantly reduce the performance variance between different SoE configurations.
> However, loading additional models increases GPU memory usage proportionally. Therefore, we must balance the trade-off between policy stability and memory constraints. For the current version, we have set the expert pool size to two.
>
> We agree that a larger expert pool holds the potential for even greater performance gains; however, realizing this potential likely requires a more complex switching strategy than simple alternation. The primary goal of this work is to introduce a new dimension for scaling model performance. The fact that our proposed simple switching strategy yields significant performance improvements effectively demonstrates the potential of this new dimension.
> We have earmarked the exploration of larger expert pools combined with advanced switching strategies as a direction for our future research.
>
> **Relevant modifications to paper:** Adding New Paragraph "Is More Experts Better" in Section 5.3.
>
> ## Weakness 4
> Thank you for your suggestion. You are correct: SoE indeed multiplies the video memory consumption. We have added a discussion in the Section 5.2 to explicitly address this trade-off. We clarify that while SoE does not increase the computational cost during testing, it results in a multiplied increase in the memory footprint (scaling with the number of loaded models).
>
> **Relevant modifications to paper:** Discussion about memory cost in "Is More Experts Better" of Section 5.3; Discussion about inference latency in "Quantitative Results" of Section 5.2.

---

> ### Author Response · Authors · 2025-12-03
> **Additional Rebuttal by Authors**
>
> ## Weakness 5
> Thank you for your valuable suggestion regarding the robustness of our offline expert pair selection.
> We want to first clarify the relevant background concerning the nuPlan dataset used in this study.
> The canonical usage dictates using Val-14 as the validation set and the nuPlan Online Test as the true test set. However, as the Online Test is currently unavailable, and given that prior work [1] suggested Val-14 provides scores similar to the online results, it has become common practice in the community to use Val-14 for both model selection (validation) and final reporting (testing).
> We acknowledge this introduces a potential data leakage concern.
> To rigorously address this issue, we introduced the Train-14 scenario set.
> It has an identical number and type of scenarios as Val-14.
> Crucially, its testing scenarios are drawn exclusively from the training dataset, ensuring zero scenario overlap with Val-14.
> Furthermore, these scenarios were generated randomly using a one-shot process, eliminating manual selection bias.We then performed our selection and testing using the standard split: Train-14 as the Validation Set and Val-14 as the Test Set.
>
> Analysis of Selection Robustness (Table 4):
>
> - General Performance (Table 4a):
> We observe that the performance improvement provided by SoE is overwhelmingly positive, meaning that SoE is almost always stronger than its constituent expert policies. However, the magnitude of the improvement varies, necessitating the use of a validation set to select the optimal SoE strategy.
>
> - Robustness of Optimal Selection (Table 4b):
>     1. In Table 4b, we use color-coding to visualize performance relative to the average score (deeper color indicates greater improvement or degradation).
>     2. We observe that the optimal pair selected on the Train-14 (Validation) set remains the optimal pair on the Val-14 (Test) set.
>     3. Furthermore, the distribution of scores achieved by SoE across the two scenario sets is highly similar.
>
> - Conclusion on Robustness: Our results demonstrate that if the validation and test distributions are generally similar, the SoE strategy chosen via the validation set is indeed robust and likely to be the best performing on the unseen test set.
>
> **Relevant modifications to paper:** Adding New Paragraph "The Robustness of SoE Selection" in Section 5.3.
>
> ## Reference
> [1]. Daniel Dauner, Marcel Hallgarten, Andreas Geiger, and Kashyap Chitta. Parting with misconceptions
> about learning-based vehicle motion planning. In Conference on Robot Learning, pp. 1268–1281.
> PMLR, 2023.

---

### Meta-Review · Area_Chair_CPku · 2026-01-07

**Summary:**

The paper received consistently negative ratings from four reviewers. The major concerns from them include the problematic design choices, the unreasonable evaluation protocol, and insufficient experimental verification. Based on the overall scores and the comments, AC decided to recommend a rejection of the paper this time.

**Reviewer Concerns:**

The reviewers had quite a few critical concerns regarding the submission, including (i)  the problematic design choices, (ii) the unreasonable evaluation protocol, and (iii) insufficient experimental verification. The rebuttal did not fully address these issues.

**Reviewer Scores:**

The reviewer score distribution is 2, 2, 2, 4. The reviewers did not show any intention to increase their scores after reading the rebuttal. The decision is thus straightforward.

---

### Decision · Program_Chairs · 2026-01-26

Reject